# The Ukraine–Russia War Is Deepening Food Insecurity, Unhealthy Dietary Patterns and the Lack of Dietary Diversity in Lebanon: Prevalence, Correlates and Findings from a National Cross-Sectional Study

**DOI:** 10.3390/nu14173504

**Published:** 2022-08-25

**Authors:** Nour Yazbeck, Rania Mansour, Hassan Salame, Nazih Bou Chahine, Maha Hoteit

**Affiliations:** 1Doctoral School of Sciences and Technology (DSST), Lebanese University, Hadath 6573, Lebanon; 2Faculty of Public Health, Section 1, Lebanese University, Beirut 6573, Lebanon; 3PHENOL Research Group (Public HEalth Nutrition prOgram Lebanon), Faculty of Public Health, Lebanese University, Beirut 6573, Lebanon; 4Lebanese University Nutrition Surveillance Center (LUNSC), Lebanese Food Drugs and Chemical Administrations, Lebanese University, Beirut 6573, Lebanon; 5Department of Nutrition and Food Sciences, Faculty of Arts and Sciences, Holy Spirit University of Kaslik (USEK), Jounieh P.O. Box 446, Lebanon; 6Program of Social Work, School of Social Sciences and Humanities, Doha Institute for Graduate Studies, Doha P.O. Box 200592, Qatar; 7Lebanese University Task Force, Lebanese University, Beirut 6573, Lebanon; 8Faculty of Pharmacy, Lebanese University, Beirut 6573, Lebanon; 9Lebanese Food, Drugs and Chemical Administration, Lebanese University, Beirut 6573, Lebanon; 10Department of Vascular Surgery, Zahraa Hospital, University Medical Center, Beirut 0961, Lebanon

**Keywords:** Russia–Ukraine war, food insecurity, Lebanon, households, dietary patterns

## Abstract

Background: Due to Russia and Ukraine’s key roles in supplying cereals and oilseeds, the Russia–Ukraine war intensifies the current food availability and price challenges in Lebanon, which is a major wheat importer. Given these constraints, we conducted this study to assess the prevalence and correlates of food insecurity, low dietary diversity (DD), unhealthy dietary patterns, and the shifts in households’ food-related habits in response to the Russia–Ukraine war among a representative sample of Lebanese household’s members aged 18 years and above (N = 914). Methods: Data were collected between June and July 2022 using self-administered questionnaires; Results: Findings showed that nearly half of the households consume an undiversified diet (46%), and 55.3% ate fewer than two meals per day. The prevalence of food insecurity among Lebanese households was 74%, with one in every four households being severely food insecure. In addition, the majority of households’ members went out shopping and purchased food less than the pre-war period (68.7% and 70.3%, respectively). Furthermore, almost 68.3% of households’ members highlighted price increases for cereal products, which were the least available and most stocked items. Findings obtained through binary logistic regression also showed that food insecurity was two times higher among households with low monthly income, 35% higher among females, and three times higher among married participants; Conclusions: The impact of the Russia–Ukraine conflict on food security in Lebanon requires a systems-thinking approach as well as international effort to understand the challenges and find solutions to minimize the war’s negative effects.

## 1. Introduction

Russia and Ukraine are two of the world’s major suppliers of agricultural commodities [1]. Prior to the Russia–Ukraine war, these two countries provided 30% of the world’s wheat and one-fifth of maize exports [1]. They also provide the vast majority of global sunflower seed product exports (80%) [1]. Furthermore, in 2021, Russia was considered among the top exporters of fertilizers, for which the prices have been rising since late 2020 as a result of higher energy prices and transportation costs associated with the COVID-19 pandemic [2]. According to the Food and Agriculture Organization (FAO), at least 50 countries rely on Russia and Ukraine for at least one-third of their wheat imports [2]. Distressingly, the Russia–Ukraine war, which began on 24 February 2022, has led to a major and deteriorating food security crisis, as well as disruptions to livelihoods in Ukraine during the agricultural growing season [2]. Although early production prospects for recently produced crops in Ukraine were promising, the war is preventing many farmers from harvesting and exporting their crops [3]. According to current reports, 20% to 30% of the crops may remain unharvested during the 2022/23 season [3]. However, yields are expected to be impacted as well [3]. In that context, this war will have many consequences on global markets and food security, posing an additional challenge to many countries, particularly low-income food imports-reliant countries and vulnerable population groups. According to recent data, war exacerbated inflationary pressures to already-high food prices caused by COVID-19 interruptions, regional weather occurrences, currency devaluations, and growing fiscal restrictions [4]. The FAO Cereal Price Index averaged 170.1 points in March 2022, which is considered to be the highest level reached during the previous 30 years [3]. As a result, increasing food prices will affect poor households especially hard, and they will be more likely to be driven further into poverty to avoid starvation [4]. Furthermore, the poorest households spend 54 percent of their consumption expenditures on food, which may force them to skip meals and consume fewer calories because they are obliged to spend a larger part of their earnings on food [4].

Lebanon, now in its third year of a catastrophic economic crisis, is confronted with a unique set of challenges that have serious implications for food security. The economic situation, political unrest, and the Beirut Port Explosions on August 4th, which resulted in the partial loss of the city’s port’s silos, altogether increased the number of Lebanese households experiencing poverty and food insecurity [5]. These complex economic crises resulted in one of the world’s ten worst economic crashes since the 1850s [5]. A recent study on food insecurity among Lebanese households conducted by Hoteit, M. et al. (2021) showed that more than half of the Lebanese population had poor dietary diversity and ate fewer than two meals per day [6]. Furthermore, the World Food Programme (WFP) estimated that one-third of the Lebanese population will be food insecure by the end of September 2021 due to the continuous economic slump [7]. Moreover, Lebanon is highly reliant on food imports, importing the majority of its wheat from Ukraine and Russia (78%), and is ranked among first ten countries in terms of imports from Ukraine [4,8]. At the same time, despite great weather conditions for agricultural output over the last two years, the financial crisis has limited farmers’ agricultural purchasing capacity [3]. As a result, national cereal production in 2021 is predicted to be lower than the that during the previous five years [3]. However, wheat accounts for 38 percent of the total caloric intake among the Lebanese population [9]. Consequently, Lebanon’s poorest households will be even less able to meet their basic demands. In the context of many issues confronting food security in Lebanon, such as the economic crisis, the COVID-19 pandemic, and recently the Russia–Ukraine war, we must indicate the causes and obstacles behind food insecurity among Lebanese families.

To date, no study had explored the impact of Russia–Ukraine war on food security among Lebanese households. Therefore, the objective of this study is to assess the prevalence and correlates of food insecurity, low dietary diversity (DD), unhealthy dietary patterns, and the shifts in households’ food-related habits in response to the Russia–Ukraine war.

## 2. Materials and Methods

### 2.1. Study Design and Sampling

The current investigation was a national cross-sectional study conducted between June and July 2022. Study participants were sampled using the probability cluster sampling technique. Subsequently, the clusters from where participants had been recruited are the eight Lebanese governorates (Mount Lebanon, Beirut, South Lebanon, North Lebanon, Akkar, Beqaa, Baalbeck-Hermel, and Nabatieh). The heads of households were recruited from each district using a probability proportional to size sampling technique. A single population formula (n = [p (1 − p)] × $[{\left(Z_{\propto /2}\right)}^{2}$/($e{)}^{2}$]) was used to determine the sample size, where n denotes the sample size, *Z*_(∝/2) is the reliability coefficient of standard error at a 5% level of significance = 1.96, p represents the probability of adults (18–64 years) who were unable to practice preventive measures of the diseases (50%), and e refers to the level of standard error tolerated (5%) as stated by Hosmer and Lemeshow [10]. Based on this formula, it was determined that the minimum acceptable sample size of 450 respondents is sufficient to ensure appropriate power for statistical analyses. Subsequently, considering a non-response rate of around 10%, we reached a total of 914 Lebanese household’s members who were invited to participate in the study by filling out an online form of a self-administered questionnaire, which was distributed through social media platforms (WhatsApp, Facebook, and Instagram). Thus, we reached a representative sample of 914 Lebanese household members for their data to be included in the analysis. The sample representativeness was enhanced by a weighting the governorate and gender variables. All household’s members voluntarily participated in the study. The online questionnaire was available in Arabic, the native language used in Lebanon. Only one household’s member from each Lebanese household aged 18 to 64 years old was considered eligible to participate in the study. On the other hand, non-Lebanese, members from the same household, and those who were not among the required age were excluded. The recruitment process is illustrated in Figure 1.

### 2.2. Study Instrument

A pre-tested, self-administered questionnaire encompassing 4 main sections and composed of different types of questions (single and multiple-choice options) was used to meet study aims. The demographic and socio-economic characteristics of the recruited households were captured in the first section. It included age, gender, governorate, marital status, education level, work discipline, history of chronic diseases, monthly income (the respondent was asked to rate his/her income compared to other Lebanese households), average monthly expenditure for food at home, and household composition. Body weight and height were self-reported by households’ members in order to allow the calculation of body mass index (BMI) [11]. In addition, the total number of persons and rooms per household were obtained to calculate the crowding index. The latter was calculated as the total number of household members (excluding infants) divided by the total number of rooms in a household (excluding kitchens and bathrooms) [12]. Households were then classified as having one person per room (no crowding), 1.5 persons per room (crowding), and more than 1.5 persons per room (over-crowding) [13]. This index was used as a proxy measure of household socio-economic status in the present study and provided a reliable result among the Lebanese population [12,14]. The second part consisted of 10 questions concerning the impact of the Russia–Ukraine War on food-related habits, including food purchasing behaviors, food consumption habits, and food storage. These questions were derived from a previously published paper by other authors [15] and by our research group describing the same study variables before the Russia–Ukraine war. It asked about the changes in dietary patterns and in the consumption data by households during the current war period in comparison to the preceding one. In the third section (10 questions), the Arab Family Food Security Scale (AFFS) was used to assess food insecurity among households in a snapshot way [16]. Using AFFS, the food security status of the Lebanese households was grouped into 3 categories (food secure, moderately food insecure, and severely food insecure). However, “moderately food insecure” and “severely food insecure” were combined under one “food insecure” category, for statistical analysis purposes. The fourth section was added to evaluate the current dietary diversity (DD) of Lebanese households by calculating the Food Consumption Score (FSC). FCS was calculated using the frequency of consumption of different food groups by a household during the seven days preceding the survey. The calculation formula of the FCS is as follows: (starches × 2) + (pulses × 3) + vegetables + fruit + (meat × 4) + (dairy products × 4) + (fats × 0.5) + (sugar × 0.5) [17]. Households were then classified as having a high DD (FCS ≥ 42) or a low DD (FCS < 42) [17]. This questionnaire is available upon request.

### 2.3. Ethical Considerations

The study was approved by the Ethical Committee at Al-Zahraa University Medical Center (#15/2022), and it follows the criteria approved by the Declaration of Helsinki. All participants provided written informed consent before taking part in the study. The participation was voluntary and exposed participants to no possible risks. On average, it takes the respondent 7–10 min to complete the survey.

### 2.4. Statistical Analysis

Data were analyzed using Statistical Package of Social Science Software (SPSS), version 25.0 (IBM, Chicago, IL, USA). A descriptive analysis was conducted, where continuous variables were reported as mean (SD) and categorical variables were reported as frequencies (N) and percentages (%). The chi-squared test was used to determine the association between categorical variables. Additionally, significant predictors of household food insecurity were determined using binary logistic regression analysis. The binary logistic regression model shows the variability of the dependent variable, which is the household food insecurity in response to the introduced independent variables. Initially, 14 variables were introduced to the model. A confidence interval of 95% was applied, and the level of significance was predetermined at 5% (*p* < 0.05 was considered to be significant).

## 3. Results

### 3.1. Demographic and Socio-Economic Characteristics of the Sampled Households

A total number of 1004 participants were screened to be included in the study. The non-response rate was 10%. A final sample of 914 participants belonging to Lebanese households was recruited. Females represented 53.4% of the study population, while 46.6% were males. The average mean age of the overall sample population was 32.0 (SD = 12.0), males (mean = 34.0; SD = 13.0), and females (mean = 31.0; SD = 11.0). Around 60.3% of participants were aged more than 24 years old. In total, 34.3% of males were between 18–24 years old, while 44.4% of females were in this age group, *p* = 0.002. Nearly half of participants (48.9%) had normal body weight, 30.9% were overweight, 16.6% were obese, and 3.6% were underweight. However, the proportion of male participants with normal BMI (41%) was significantly lower than the proportion of females with acceptable body weight (55.8%). Moreover, the proportion of males with overweight (38.2%), and obesity (19.4%) was higher than females (24.6% and 14.1%, respectively), *p* < 0.001. Furthermore, 13.6% of the households were residing in Mount Lebanon, 13.1% in Beirut, 12.6% in North Lebanon, and 11.4% in South Lebanon. Moreover, 13% and 11.8% lived in the Bekaa and Baalbeck-Hermel, respectively. In addition, more than half of participants (51.2%) were single, 45% were married, while only 1.8%% were divorced. However, the proportion of single females (55.2%) was significantly higher than that of males (46.6%), *p* = 0.002. Regarding their educational level, the majority (74.4%, n = 680) of study participants had university education level, whereas only 1% reported that they are illiterate. Concerning the working status, the majority of respondents (40.4%) reported that they work, 28.4% were students, while 26.1% were not working. The proportion of unemployed females (29.6%) was significantly higher than that of males (22.1%), *p* < 0.001. As for chronic diseases, 32.1% of the study population reported that they had a history of chronic diseases. However, an equal proportion of males and females reported having chronic diseases, *p* = 0.071. Among the 294 sample participants who had chronic diseases, 33% had depression, followed by hypertension (25%, n =74) and diabetes (21.42%, n = 63). Other diseases were also reported, including asthma (19.38%, n = 57), cardiovascular diseases (17%, n = 50), osteoporosis (11.9%, n = 35), liver diseases (6.46%, n = 19), renal diseases (4.76%, n = 14), and anemia (2.72%, n = 8). Moreover, diabetes was the most reported disease by males; however, depression was mostly reported by females, *p* < 0.001 (Data not shown). Concerning the monthly income, 41.8% of households were earning more than 1.5 million LBP and 17.5% were earning less than 1.5 million LBP. Furthermore, the current household income was less than 300 USD for 19.8%, and more than 300 USD for 14%. However, lower monthly income (<1.5 million LBP) was reported more by females (20.4%), as compared to males (14.2%), *p* = 0.002. However, around 41.8% of the households reported that they spent more than 3 million LBP as an average monthly expenditure for food at home. In terms of the household characteristics of the study sample, most of the households were composed of multiple adults (45.6%), 33.9% were composed of multiple adults with at least one child, while 10.4% and 10.1% were composed of one adult with at least one child and one adult, respectively. Furthermore, nearly half of the participants (56.5%) reported that they do not have children, 32.7% have less than three children, and only 10.8% reported that they have more than three children. In addition, 49.6 % had a crowding index of ≤1 person/room (no crowding), whereas 28.4% of respondents had an over-crowding index (>1.5 persons/room). Moreover, nearly half of the household heads of the participants were aged more than 50 years old (52.7%), 38.2% were between 35–50 years old, and only 9% were less than 35 years old. Additionally, 33.4% of household heads were studying or had studied at university, while only 6.5% were illiterate. To investigate the impact of the Russia–Ukraine war on the economic situation in Lebanon, households were asked about the impact of this war on their monthly income, where 34% admitted that their income had declined after the Russia–Ukraine war. Moreover, nearly half the households reported that their monthly income is estimated to be less than most other Lebanese households (48%). These findings are presented in Table 1.

### 3.2. Indicators of Household Food Security

#### 3.2.1. Households’ Dietary Diversity (DD)

In the present study, 55.3% of households reported to consume two meals or fewer in a day, with 64.3% describing this as a typical meal pattern. The study findings reveal that, among all food groups, cereals were the most frequently consumed 4 days or more, as reported by more than half the households (55.3%). Nonetheless, most households reported consuming white tubers (71.3%), vegetables (63.7%), fruits (72.8%), eggs (87.0%), pulses and nuts (84.0%), dairy products (78.8%), fats and oils (69.8%), sweets (69.8%), and spices and condiments (67.3%) in 3 days or fewer. More females than males reported a 3-day or fewer consumption of pulses and nuts (88.4% vs. 78.9%, *p* < 0.001), fats and oils (73.5% vs. 65.7%, *p* = 0.01), and sweets (73.0% vs. 66.2%, *p* = 0.027) (Table 2).

Hence, 46% of the Lebanese households were observed to consume an undiversified diet (Figure 2. Of interest, Akkar district had the largest proportion of households having low dietary diversity (79.1%), followed by North Lebanon (67.7%), Beqaa (54.3%), Baalbeck-Hermel (40.4%), South Lebanon (39.1%), Beirut (36.3%), Mount Lebanon (35.7%), and Nabatieh (18.5%) (Figure 2). Furthermore, households with low dietary diversity were relying mainly on cereals and white tubers. However, fruit, pulses, meat, and dairy products were less consumed compared to those who have high dietary diversity (data not shown).

#### 3.2.2. Household Food Security Scale

Using the scale Arab Family Food Security Scale (AFFSS), the majority (74%) of households were food-insecure (including “moderately food insecure” and “severely food insecure”) (Figure 3a). Additionally, among those who were food-insecure, 29% of households were severely food insecure (Figure 3a). However, the highest percentage of food insecurity (“moderately food insecure” and “severely food insecure) was observed in Akkar and North Lebanon (86%) (Figure 3b). Interestingly, the highest percentage of food security was in Beirut (50%) (Figure 3b).

### 3.3. Changes in Food Shopping Behaviors during the Outbreak of Russia–Ukraine War

Regarding shopping behaviors, in comparison with the pre-war period, 68.7% of the household’s members stated making fewer grocery-shopping trips than usual after the outbreak of the Russia–Ukraine war, 30.1% reported going shopping as usual, while only 1.3% stated that they go shopping more than usual. The percentage of females who reported going shopping less than usual was significantly higher than that of males (73.1% and 63.6%, respectively), *p* = 0.009. Additionally, the majority of household’s members (70.3%) specified that they buy less food than usual on each shopping trip. Moreover, a higher significant proportion of females (74.6%) mentioned buying food less than usual, as compared to males (65.3%), after the outbreak of the Russia–Ukraine war, *p* = 0.026. Furthermore, almost 70.3% of the households reported that their food wastage decreased after the Russia–Ukraine war. Moreover, 35.6% of the households reported having stocked up food during the war (Table 3).

The findings also indicated that the most stocked items during the Russia–Ukraine war were cereal products (55% of the households), oils (22.1%), sugars (16.4%), and legumes (14.6%). Additionally, the results revealed that some food items were less available after the outbreak of the Russia–Ukraine war compared to the pre-war period, including cereals products (55.3% of the households), oils (35.1%), and sugars (19.9%). However, the majority of households (68.3%) highlighted price increase for cereals products. Other items were also highlighted for their price increase including oils (51.7%), legumes (46.2%), roots and tubers (46.2%), milk and dairy products (35.2%), fruits and vegetables (34.8%), and canned food (33.2%) (Table 4).

As shown in Figure 4, in comparison to the pre-war period, the findings of food sourcing during the Russia–Ukraine war demonstrate significant changes in Lebanese households’ behaviors related to food shopping. Specifically, 28.5% of the household’s members ate out less (e.g., restaurants/cafeteria/fast-food) and 27.4% ordered less take-away or fast-food meals with deliveries (all by including “slightly less” and “much less” answer options). In addition, the majority of household’s members stated that they never order groceries or meals online (69.8% and 60.2%, respectively). On the other hand, a significant proportion of the household’s members buy less food in person from large supermarkets (41.4%) or small supermarkets (44.5%) during the Russia–Ukraine war (all by including “slightly less” and “much less”) (Figure 4).

### 3.4. Changes in Food Consumption Behaviors during the Russia–Ukraine War

When asking household’s members about food consumption patterns during the current situation in comparison to the preceding time, the results showed that 67.1% of Lebanese households consumed less meat, 61.1% consumed fewer fruits and vegetables, 53.9% fewer sweets, cookies, cakes and candies, 50.1% fewer healthy snacks, and 44.3% less healthy food compared to the period preceding the war (Figure 5).

### 3.5. Correlates of Household Food Insecurity

Table 5 shows the relationship between the food insecurity and households’ characteristics. The proportion of food insecurity was significantly higher among those aged more than 24 years old than those aged 18 to 24 years old (66.1% and 33.9%, respectively), *p* < 0.001. However, food insecurity prevalence was significantly higher among households having household heads older than 50 years old (49.9%) compared to households with younger household heads, *p* = 0.005. Additionally, the prevalence of food insecurity was significantly higher among females than males (56.5% and 43.5%, respectively), *p* = 0.001. Although participants with normal BMI had the highest percentage of food insecurity (49.6%), the difference was not significant compared to overweight (31.0%), obese (16.1%), and underweight (3.3%) participants, *p* = 0.628. Furthermore, 14.5% of food-insecure households were residing in North Lebanon, followed by Mount Lebanon (14.1%), Beqaa (13.8%), and Akkar (13.5%), *p* < 0.001. Among food-insecure households’ members, nearly half (50.9%) were married, which was significantly higher than those who were single (44.3%), *p* < 0.001. Similarly, respondents who reported that they studied or were studying at university had the highest proportion of food insecurity (67.3%), *p* < 0.001. However, household heads who had school education level were shown to have the highest proportion of food insecurity (64.7%), *p* < 0.001. Moreover, the proportion of food insecurity among unemployed households’ members (30.3%) was significantly higher than employed household’s members (38.6%), *p* < 0.001. However, non-medical sector workers had a significantly higher proportion of food insecurity, as opposed to medical sector workers (86.3% versus 13.7%, *p* < 0.001). Households with more than three children (4.9%) were shown to have a significantly lower proportion of food security compared to households with fewer children, *p* < 0.001. Additionally, food insecurity prevalence was the highest among households composed of multiple adults (43%), *p* = 0.001. Households earning the highest income (>300 USD/month) had the highest proportion of food security (38.2%), *p* < 0.001. Moreover, the proportion of food-insecure households who reported a decline in their monthly income (22.5%) after the Russia–Ukraine war was significantly higher than the proportion of households who reported an increase in their monthly income (0.3%), *p* < 0.001. Moreover, food insecurity prevalence was the highest among households spending 1 to 3 million LBP as an average monthly expenditure for food (38.7%), *p* < 0.001. The percentage of food-insecure households with over-crowding (32.5%) was shown to be considerably more than food-insecure crowded households (22.5%), *p* < 0.001. Interestingly, among food-insecure households, the proportion of households with low DD (55.2%) was significantly higher than that for those with high DD (44.8%), *p* < 0.001.

### 3.6. Determinants of the Household Food Insecurity: Binary Logistic Regression Analysis

According to Table 6, many study variables were affecting food security of the Lebanese households. including the following: gender, marital status, residence, weight status, job nature, education level of the household head, and monthly income. The backward stepwise analysis shows that females compared to males had 35% higher probability to be food insecure (OR = 0.656; 95% CI (0.450–0.958), *p* = 0.029). Moreover, married respondents were 2.9 times more likely to be food insecure compared to single respondents (OR = 2.989; 95% CI (1.944–4.597), *p* < 0.001). The residency was shown to be an additional determinant; the highest estimated probability of food insecurity was among households residing in Mount Lebanon (OR = 3.393, 95% CI (1.768–6.510), *p* = <0.001) compared to those residing in Beirut. Furthermore, underweight household members had an 18% higher probability of being food insecure than normal-weight participants (OR = 0.821, 95% CI (0.322–2.098), *p* = 0.681). Additionally, the likelihood of food insecurity is significantly higher among those who worked in the non-medical sector (OR = 1.598, 95% CI (1.032–2.473), *p* = 0.036) compared to those who worked in the medical sector. Moreover, households with illiterate household heads were predicted to be food insecure by 52% compared to household heads with a university education level (OR = 0.481; 95% CI (0.179–1.294), *p* = 0.147). The backward analysis also shows that households having no monthly income were 90% more likely to be food insecure in contrast to those having an income of more than 300 USD (OR = 0.096, 95% CI (0.032–0.284), *p* < 0.001).

## 4. Discussion

This study explored the impact of the Russia–Ukraine war on food security among Lebanese households. It also examined the changes in food-related habits in Lebanese households. Overall, amid the Russia–Ukraine war, the majority of households’ members went out shopping and purchased food less than usual (68.7% and 70.3%, respectively). In addition, almost 68.3% of households highlighted price increases for cereal products, which were the least available and most stocked items. The overall prevalence of food insecurity among Lebanese households was 74%, with one in every four households being severely food insecure. Moreover, nearly half of the households were consuming undiversified diets (46%) and 55.3% ate fewer than two meals per day.

In the current study, 68.7% of households reported making fewer shopping trips than usual since the outbreak of the Russia–Ukraine war. Furthermore, the majority of households stated that they purchase less food than usual. This could be related to the economic instability in Lebanon, which derives from many factors, including the Russia–Ukraine war, financial crisis, the Beirut Port Explosions on 4 August, and the COVID-19 pandemic. To facilitate, consumer purchasing behavior is believed to be the total of decision-making processes influenced by both internal and external factors [10]. Economic insecurity is the most critical external influence. Economic crises have a negative effect on both planned and unplanned purchasing behaviors of all consumers [18]. According to Ahorsu et al., 2020, when consumers feel insecure and anxious, their willingness to consume decreases [19]. Moreover, if consumer confidence declines as a result of economic uncertainty, expenditure budgets are considerably lower. Thus, consumers limit their future consumption by shopping and purchasing less frequently [20]. Consequently, the absence of panic-buying resulted in decreased food waste. Study findings showed that the majority of households (70.3%) reported that their food wastage decreased during the outbreak of the Russia–Ukraine war. Hopefully, this shows a potential path toward a more sustainable behavior in food consumption. Similarly, the COVID-19 pandemic has resulted in less food waste in Lebanon [21], as well as in other countries, including Qatar [15], Tunisia [22], Morocco [23], and the US [24]. However, only 35.6% of households stated that they stock up food, and this can be due to food-price inflation. In Lebanon, the cost of the basic food basket increased by 1,140 percent by January 2022 compared to October 2019, affecting household finances for basic requirements and preventing people from being able to store food [25]. This also explains our findings, in which households reported lower consumption for nearly all food groups. However, the least consumed food items were meat (67.1%) and fruits and vegetables (61.1%), indicating that households limit their purchases of food types they cannot afford, such as meat and fish, and begin consuming larger quantities of cheaper food types. Interestingly, findings also showed that the majority of households (68.3%) highlighted price increase for cereals products. Moreover, results indicate an increase in the purchase as well as the storage of non-perishable food items including cereal products, oils, and sugar. Consequently, the availability of these items was affected, where nearly half of households (55.3%) in our study stated that cereals products and oils were the most unavailable item. These findings are unsurprising, given that the Russia–Ukraine conflict exposed global markets to increased risks of shortages and international price inflations [1]. Russia and Ukraine are two of the world’s major agricultural commodity producers [1]. Prior to the crisis, the two countries supplied 30% of the world’s wheat and one-fifth of maize exports [1]. They also accounted for the majority of the global sunflower seed product exports (80%) [1]. Lebanon, like other nations in the Middle East and North Africa, relies largely on wheat imports from Russia and Ukraine [26]. However, the Russia–Ukraine war limited wheat access for most wheat importers [9]. Since wheat is a vital staple food throughout the Middle East and North Africa, disruptions in the wheat supply chain generated critical food security challenges in the region [26]. Consequently, the food Consumer Price Index (CPI) has increased significantly in most countries in the region in June 2022, including Lebanon (216%), Syria (71%), Egypt (24.2%), Morocco (9.5%), Iraq (7.6%), and Yemen (43%), compared to the same period last year [26]. Moreover, local cereal production in Lebanon accounts for less than 20% of total consumption [4,7]. To illustrate, during the past two years, the economic crisis has hindered the agricultural capacity of Lebanese farmers [7]. Many farmers prefer to take farming as a secondary activity, especially in places with greater economic opportunities [7]. In addition, following the explosion of Beirut’s port in 2020 and the partial loss of the city’s port’s silos, seed storage became extremely difficult [9]. Thus, the Russia–Ukraine war limited access to wheat for import-dependent Lebanon, which was already in the grip of one of the world’s worst economic crises since the mid-nineteenth century, and, in addition to wheat, other consumer products from Ukraine and Russia, such as cooking oil and milk powder, were missing from supermarket shelves [9].

Our findings also show that nearly half of the households were consuming undiversified diets (46%), and 55.3% ate fewer than two meals per day. These findings complement a recent study that aimed to assess the impact of crises on food security in Lebanon, where it showed also that 53% of households had poor DD and 55.8% ate fewer than two meals per day [6]. However, our findings were higher than those reported in previous studies among other countries in the MENA region with middle-income, such as Jordan (23.8%) and Palestine (14%) [27,28]. Additionally, Akkar is found to have the largest proportion of households with a low DD (79.1%). This result came hand-in-hand with previous reports, where Akkar presented the poorest DD compared to other governorates [29]. Furthermore, the latter results were corroborated by another study in Lebanon that showed that Beqaa and Akkar had the highest proportions of households with low DD (83% and 73%, respectively) [6].

Using the Arab Family Food Security Scale (AFFSS), three out of four households were food insecure. Similarly, a recent study done in 2021 showed that almost 75.4% of Lebanese households were food insecure [6]. In a country which is heavily dependent on food imports like Lebanon, and which is already suffering from major economic crisis, the Russia–Ukraine war will intensify the current food availability and price challenges. Reduction of crop supply resulting from the military conflict in Ukraine have lowered the available crops for food consumption. According to the Food and Agriculture Organization of the United Nations (FAO), the pillars of food security include availability, access, utilization, and stability [30]. However, the “availability” pillar of food security is affected by the shortage of food and crops supply, leading also to food insecurity [31]. At the governmental level, the highest percentage of food insecurity was among Akkar and North Lebanon (86%). According to recent reports, Akkar also recorded the highest proportion of food shortages (71%) followed by North Lebanon (63%) [32]. The fact that a high proportion of the Lebanese population are food insecure is important by itself, but the potential negative short-term and long-term health effects of food insecurity are potentially more concerning. Some of the studies have shown that food insecurity is associated with decreased nutrient intakes [33,34], increased rates of mental health problems and depression [35,36], diabetes [37,38], cardiovascular diseases [39], hyperlipidemia [38], and poor sleep outcomes [40]. In our study, the adjusted binary logistic regression analysis showed that households with low monthly income (less than 1.5 million LBP) and illiterate household heads had higher probability of food insecurity. These findings are rational, as the indicators of low socio-economic status such as lower education, low income, fewer asserts, and unemployment have been frequently associated with food insecurity [6,41,42].

### Strengths and Limitations

The current study has several strengths. It is the first study in Lebanon to investigate the prevalence and correlates of household food insecurity in the context of the Russia–Ukraine conflict. Another strength is the use of a regionally validated household food security access scale. On the other hand, this study has several limitations that should be considered in order to improve the applicability of our findings. The survey’s cross-sectional design limits the ability to draw causal conclusions. Furthermore, the online survey excluded a particularly vulnerable population that is difficult to track down through social media platforms, leading to a possible selection bias. Another limitation is that food insecurity was measured at the household level, and it may not assess food security status at the individual level. In addition, it is important to mention that Lebanon was already affected by severe economic crisis before the outbreak of the Russia–Ukraine war, which might bias some of our findings.

## 5. Conclusions

In conclusion, the current study’s findings revealed that the prevalence of food insecurity was remarkably high among Lebanese households. However, these findings should dispel any lingering concerns that Lebanon’s government and non-governmental organizations are reverting their efforts to reduce hunger and food insecurity. As a result, in the absence of a major multifaced transformation, Lebanon’s complex crisis is expected to persist in 2023, raising concerns about food insecurity consequences across multiple demographic groups. However, the additional impact of the Russia–Ukraine war on food security in Lebanon highlights the importance of a systems-thinking approach and international action to better understand the challenges and develop strategies to reduce the war’s terrible effects. Hopefully, this study would offer baseline data to all relevant organizations, contributing to the development of an evidence-based approach for food security interventions among the Lebanese community.

## Figures and Tables

**Figure 1 nutrients-14-03504-f001:**
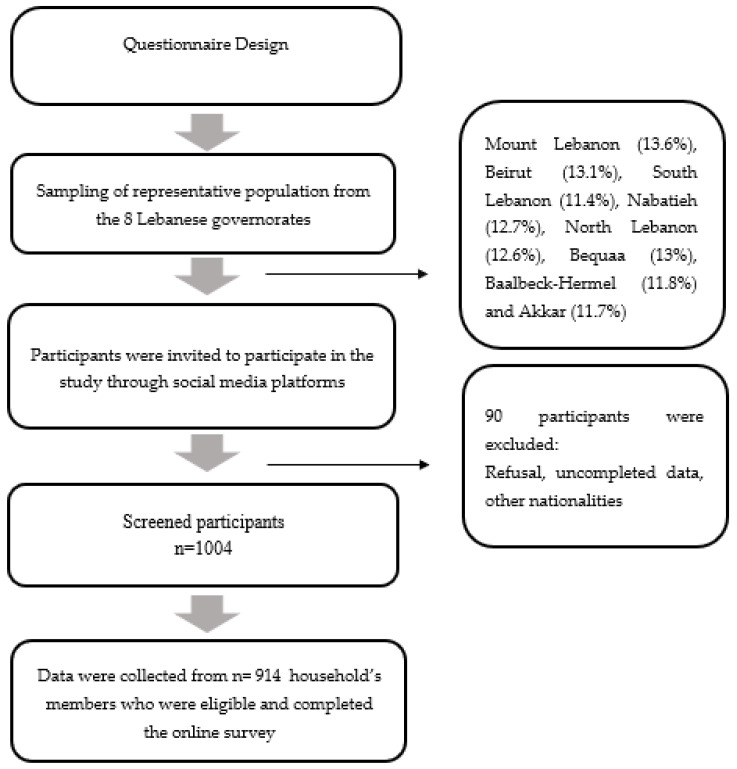
Flow Chart of the Recruitment Process in the Study.

**Figure 2 nutrients-14-03504-f002:**
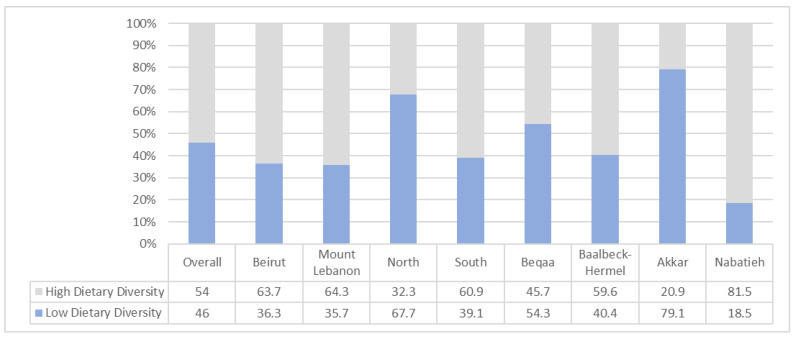
Households’ dietary diversity, overall and by governorate.

**Figure 3 nutrients-14-03504-f003:**
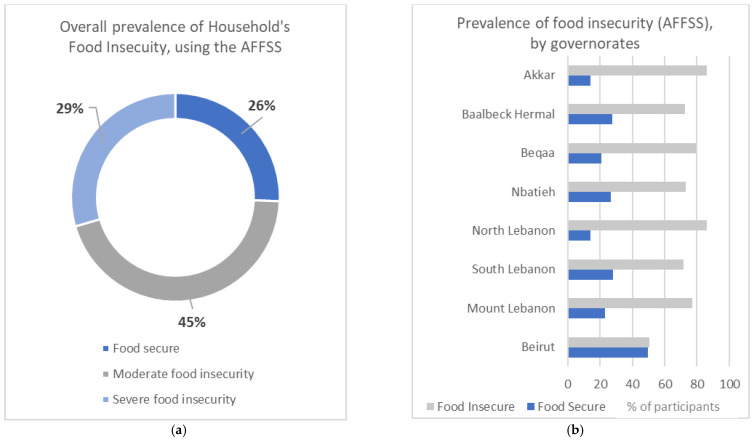
(**a**) Overall prevalence of household’s food insecurity. (**b**) Prevalence of household’s food insecurity, by governorate.

**Figure 4 nutrients-14-03504-f004:**
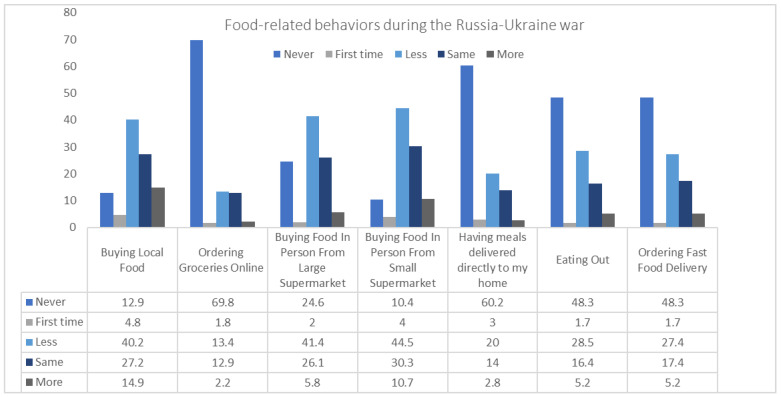
Changes in food-related behaviors during the Russia–Ukraine war.

**Figure 5 nutrients-14-03504-f005:**
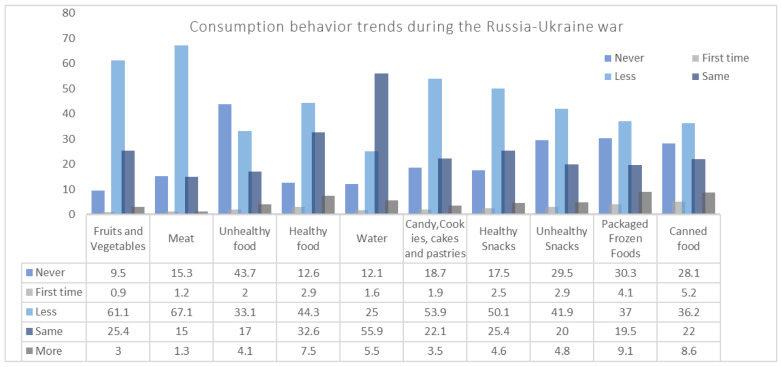
Changes in consumption behavior trends during the Russia–Ukraine war.

**Table 1 nutrients-14-03504-t001:** Demographic and socio-economic characteristics of the sampled households, overall and by gender.

		Overall (N = 914)		Males (N = 426)		Females (N = 488)		
		Mean	SD	Mean	SD	Mean	SD	
Age in years		**32.0**	12.0	34.0	13.0	31.0	11.0	
		N	%	N	%	N	%	*p*-value
Age Categories	18–24	364	39.7	146	34.3	218	44.4	**0.002**
	>24	550	60.3	279	65.7	271	55.6	
BMI Classification	Underweight	33	3.6	6	1.4	27	5.5	**<0.001**
	Normal	446	48.9	174	41.0	272	55.8	
	Overweight	284	30.9	163	38.2	120	24.6	
	Obese	151	16.6	83	19.4	69	14.1	
Gender	Male	426	46.6	426	100.0	0	0.0	-
	Female	488	53.4	0	0.0	488	100.0	
Residency	Beirut	120	13.1	63	14.8	57	11.7	**0.003**
	Mount Lebanon	125	13.6	71	16.7	54	11.0	
	South Lebanon	105	11.4	36	8.5	68	14.0	
	Beqaa	118	13.0	62	14.5	57	11.7	
	Baalbeck-Hermel	108	11.8	44	10.2	64	13.2	
	Akkar	107	11.7	38	9.0	68	14.0	
	Nabatieh	116	12.7	58	13.7	58	11.9	
	North Lebanon	115	12.6	54	12.6	61	12.5	
Marital Status	Single	467	51.2	198	46.6	269	55.2	**0.002**
	Married	411	45.0	217	51.1	193	39.7	
	Divorced	17	1.8	6	1.2	11	2.3	
	Widowed	19	2.1	5	1.1	14	2.9	
Education Level	Illiterate	10	1.0	5	1.3	4	0.8	0.567
	School level	224	24.5	110	25.9	114	23.4	
	University level	680	74.4	310	72.9	370	75.8	
Current Occupation	Working	369	40.4	217	50.9	152	31.2	**<0.001**
	Not Working	238	26.1	94	22.1	144	29.6	
	Student	260	28.4	104	24.3	156	32.0	
	Other	47	5.2	11	2.7	36	7.3	
Job Nature	Medical sector	162	17.7	64	15.1	97	20.0	0.055
	Non-Medical sector	752	82.3	361	84.9	390	80.0	
Household Crowding Index	No Crowding (≤1 person per room)	453	49.6%	214	50.2	239	49.0	0.917
	Crowding (1–1.5 person per room)	201	22.0%	92	21.7	109	22.3	
	Over Crowding (>1.5 person per room)	260	28.4%	119	28.1	140	28.7	
Number of children	None	516	56.5	222	52.2	294	60.3	**0.024**
	3 or less	298	32.7	147	34.7	151	30.9	
	More than three	100	10.8	57	13.2	43	8.8	
Household Composition	One adult	93	10.1	46	10.8	47	9.6	0.848
	Multiple adults	416	45.6	193	45.3	224	45.8	
	One adult with at least one child	95	10.4	41	9.6	54	11.1	
	Multiple adults with at least one child	310	33.9	147	34.3	164	33.5	
Age of Household Head	<35 years	83	9.0	48	11.3	34	7.1	0.077
	35–50 years	349	38.2	158	37.1	191	39.2	
	>50 years	482	52.7	220	51.6	262	53.7	
Household head’s Education level	Illiterate	60	6.5	18	4.3	41	8.4	**<0.001**
	School level	549	60.1	228	53.5	321	65.9	
	University	305	33.4	180	42.2	125	25.6	
Monthly Income	None	64	7.0	31	7.2	33	6.8	**0.002**
	Less than 1.5 million L.B.P.	160	17.5	60	14.2	99	20.4	
	≥1.5 million L.B.P.	382	41.8	165	38.7	217	44.5	
	≤300 USD	180	19.8	105	24.6	76	15.5	
	More than 300 USD	128	14.0	65	15.3	63	12.9	
Income status compared to other households	Less than most other Lebanese households	438	48	177	41.8	261	53.5	**0.001**
	It is not different from the income of other Lebanese households	319	34.9	170	39.9	149	30.6	
	More than the income of other Lebanese households	157	17	80	18.4	77	16	
Impact of Russia–Ukraine war on Monthly Income	My salary does not change	566	62.0	270	63.5	296	60.7	**0.009**
	My salary decreases	312	34	134	30.9	178	36.5	
	My salary increases	36	4.0	23	5.6	13	2.8	
Average Monthly Expenditure for Food at Home	Less than 675,000 LBP	35	3.7	14	3.0	21	4.3	**0.03**
	675,000–1 million LBP	144	15.8	56	13.2	88	18.1	
	1 million–3 million LBP	353	38.7	159	37.3	194	39.9	
	More than 3 million LBP	382	41.8	198	46.5	184	37.7	

Bold means significant at *p*-value < 0.05.

**Table 2 nutrients-14-03504-t002:** Food groups consumption per week in overall population and by gender.

		Overall		Male		Female		
		N	%	N	%	N	%	*p*-Value
Number of meals consumed the day before	2 meals and less	505	55.3	238	55.9	267	54.8	0.722
	3 meals and more	408	44.7	187	44.1	220	45.2	
Number of meals re ported as usual, less, or more	Less than usual	311	34.1	123	28.8	189	38.7	**0.005**
	As usual	587	64.3	297	69.8	290	59.5	
	More than usual	15	1.6	6	1.4	9	1.8	
Consumption of food groups during the previous 7 days								
Cereals	3 days or fewer	408	44.7	177	41.5	231	47.4	0.074
	4 days and more	505	55.3	249	58.5	256	52.6	
White tubers	3 days or fewer	651	71.3	297	69.9	353	72.5	0.386
	4 days and more	262	28.7	128	30.1	134	27.5	
Vegetable	3 days or fewer	582	63.7	263	61.8	319	65.4	0.274
	4 days and more	331	36.3	162	38.2	169	34.6	
Fruit	3 days or fewer	665	72.8	313	73.5	352	72.2	0.649
	4 days and more	248	27.2	113	26.5	136	27.8	
Eggs	3 days or fewer	795	87.0	364	85.6	431	88.3	0.23
	4 days and more	119	13.0	61	14.4	57	11.7	
Pulse and nuts	3 days or fewer	767	84.0	336	78.9	431	88.4	**<0.001**
	4 days and more	146	16.0	90	21.1	57	11.6	
Dairy products	3 days or fewer	720	78.8	338	79.4	382	78.4	0.695
	4 days and more	193	21.2	88	20.6	106	21.6	
Fat and oils	3 days or fewer	638	69.8	279	65.7	358	73.5	**0.010**
	4 days and more	276	30.2	146	34.3	129	26.5	
Sweets	3 days or fewer	638	69.8	282	66.2	356	73.0	**0.027**
	4 days and more	275	30.2	144	33.8	132	27.0	
Spices and condiments	3 days or fewer	615	67.3	298	69.9	317	65.0	0.108
	4 days and more	299	32.7	128	30.1	171	35.0	
Meat	3 days or fewer	735	80.5	343	80.6	392	80.3	0.886
	4 days and more	178	19.5	82	19.4	96	19.7	
Fish	3 days or fewer	887	97.1	409	96.1	477	97.9	0.121
	4 days and more	27	2.9	16	3.9	10	2.1	

Bold means significant at *p*-value <0.05.

**Table 3 nutrients-14-03504-t003:** Shopping behavior and food wastage changes during the Russia–Ukraine war.

		Overall (N = 914)		Male (N = 426)		Female (N = 488)		
		N	%	N	%	N	%	*p*-Value
Shopping behavior change	I go shopping less than usual	627	68.7	271	63.6	357	73.1	**0.009**
	I go shopping like I used to	275	30.1	149	35.0	126	25.7	
	I go shopping more than usual	12	1.3	6	1.4	6	1.2	
Change of food purchase	I buy less than usual	642	70.3	278	65.3	364	74.6	**0.026**
	I buy as same as usual	239	26.1	130	30.4	109	22.3	
	I buy a lot more than usual	33	3.6	18	4.2	15	3.1	
Food Wastage	Less	642	70.3	288	67.8	354	72.6	0.076
	Has not changed	210	23.0	112	26.2	98	20.1	
	More	61	6.7	26	6.0	36	7.3	
Stocking up food	Yes	325	35.6	153	36.0	172	35.2	0.812
	No	589	64.4	273	64.0	316	64.8	

Bold means significant at *p*-value < 0.05.

**Table 4 nutrients-14-03504-t004:** Types of foods stocked up, change in food availability, and food price increase during the Russia–Ukraine war.

	Type of Food Stocked Up		Notice of Less Available Food		Notice of Any Food Price Increase	
	N	%	N	%	N	%
Cereals and their products (bread, rice, pasta, flour, etc.)	503	55.0	505	55.3	623	68.3
Roots and tubers (potatoes, etc.)	53	5.8	95	10.5	415	45.5
Legumes (e.g., peas, chickpeas)	133	14.6	77	8.5	422	46.2
Sugar	150	16.4	181	19.9	101	11.0
Oils	201	22.1	321	35.1	472	51.7
Fruits and Vegetables	9	1.0	55	6.0	317	34.8
Meat and meat products	4	0.4	33	3.7	73	8.0
Fish and seafood	3	0.3	85	9.3	299	32.7
Milk and dairy products	34	3.8	124	13.6	321	35.2
Canned food	90	9.9	55	6.0	303	33.2
None	286	31.3	123	13.5	73	8.0

**Table 5 nutrients-14-03504-t005:** The association of demographic and socioeconomic characteristics with household food insecurity.

	Household Food Insecurity according to (AFFSS)		
	Food-Secure n (%)	Food-Insecure n (%)	*p*-Value
Age			**<0.001**
18–24	132 (56.6)	231 (33.9)	
>24	101 (43.4)	449 (66.1)	
Gender			**0.001**
Male	130 (55.6)	296 (43.5)	
Female	104 (44.4)	384 (56.5)	
Body Mass Index (BMI)			0.628
Underweight	11 (4.6)	22 (3.3)	
Normal	109 (46.8)	337 (49.6)	
Overweight	72 (30.7)	211 (31.0)	
Obese	42 (18.0)	109 (16.1)	
Residence			**<0.001**
Mount Lebanon	29 (12.2)	96 (14.1)	
Beirut	59 (25.4)	61 (8.9)	
South Lebanon	30 (12.6)	75 (11.0)	
North Lebanon	16 (6.9)	99 (14.5)	
Akkar	15 (6.4)	92 (13.5)	
Beqaa	24 (10.4)	94 (13.8)	
Baalbeck-Hermel	30 (12.7)	78 (11.5)	
Nabatieh	31 (13.3)	85 (12.5)	
Marital Status			**<0.001**
Single	166 (71.0)	301 (44.3)	
Married	65 (27.8)	346 (50.9)	
Divorced	2 (0.8)	15 (2.1)	
Widowed	1 (0.4)	18 (2.6)	
Education level			**<0.001**
Illiterate	0 (0.0)	9 (1.4)	
School level	11 (4.7)	213 (31.4)	
University level	223 (95.3)	457 (67.3)	
Current Job			**<0.001**
Working	106 (45.3)	263 (38.6)	
Not working	32 (13.8)	206 (30.3)	
Student	87 (37.4)	182 (25.3)	
Other	8 (3.5)	39 (5.7)	
Job Nature			**<0.001**
Medical Section	69 (29.4)	93 (13.7)	
Non-medical Section	165 (70.6)	587 (86.3)	
Number of children per household			**<0.001**
No children	176 (75.4)	340 (50.0)	
3 or less children	46 (19.7)	252 (37.1)	
More than 3 children	11 (4.9)	87 (12.9)	
Household Composition			**0.001**
One adult	20 (8.7)	72 (10.7)	
Multiple adults	124 (52.9)	292 (43.0)	
One adult with at least one child	10 (4.4)	85 (12.5)	
Multiple adults with at least one child	80 (34.0)	230 (33.8)	
Household Head Education level			**<0.001**
Illiterate	7 (3.0)	52 (7.7)	
School level	109 (46.8)	440 (64.7)	
University level	117 (50.2)	187 (27.6)	
Household Head Age			**0.005**
<35 years	23 (9.6)	60 (8.8)	
35–50 years	69 (29.4)	280 (41.3)	
>50 years	143 (61.0)	339 (49.9)	
Household’s Monthly Income			**<0.001**
None	5 (2.1)	59 (8.7)	
Less than 1.5 million L.B.P.	6 (2.6)	153 (22.6)	
≥1.5 million L.B.P.	80 (34.2)	302 (44.4)	
≤300 USD	54 (22.9)	127 (18.7)	
More than 300 USD	89 (38.2)	39 (5.7)	
The impact of the Russia–Ukraine war on the household’s monthly income			**<0.001**
No impact	216 (92.2)	525 (77.2)	
A decline in the monthly income	17 (7.4)	153 (22.5)	
An increase in the monthly income	2 (0.3)	2 (0.3)	
Average Monthly Expenditure for Food at Home			**<0.001**
Less than 675,000 LBP	2 (0.9)	32 (4.6)	
675,000–1 million LBP	14 (6.1)	130 (19.1)	
1 million–3 million LBP	90 (38.5)	263 (38.7)	
More than 3 million LBP	127 (54.5)	255 (37.5)	
Household Crowding Index			**<0.001**
No crowding (≤1)	147 (63.0)	306 (45.0)	
Crowding (1–1.5)	48 (20.5)	153 (22.5)	
Over-crowding (>1.5)	39 (16.5)	221 (32.5)	
Household’s Dietary Diversity (FCS)			**<0.001**
Low	45 (19.4)	375 (55.2)	
High	188 (80.6)	305 (44.8)	

Bold means significant at *p*-value < 0.05.

**Table 6 nutrients-14-03504-t006:** The determinants of Households’ Food Insecurity based on the Logistic Regression analysis (Backward stepwise method).

Determinants of Food Insecurity Food Secure vs. Food Insecure	OR	95% CI For EXP (B)		*p*-Value
		Lower	Upper	
Gender				
Female (Reference)	1.00			
Male	0.656	0.4 50	0.9 58	0.029
Marital Status				
Single (Reference)	1.00			
Married	2.989	1.944	4.597	<0.001
Divorced	2.689	0.493	14.681	0.253
Widowed	3.613	0.350	37.261	0.281
Residency				
Beirut (Reference)	1.00			
Mount Lebanon	3.393	1.768	6.510	<0.001
North Lebanon	1.715	0.802	3.668	0.164
South Lebanon	1.759	0.898	3.446	0.100
Beqaa	2.401	1.205	4.784	0.013
Baalbek Hermel	1.866	0.939	3.708	0.075
Akkar	2.055	0.921	4.585	0.079
Nabatieh	3.254	1.690	6.263	<0.001
BMI				
Normal (Reference)	1.00			
Underweight	0.821	0.322	2.098	0.681
Overweight	0.739	0.476	1.148	0.178
Obese	0.451	0.265	0.769	0.003
Job Nature				
Medical (Reference).	1.00			
Non-medical	1.598	1.032	2.473	0.036
Education Household Head				
Illiterate (Reference)	1.00			
School level	0.786	0.301	2.049	0.622
University level	0.481	0.179	1.294	0.147
Monthly Income				
None (Reference)	1.00			
Less than 1.5 million L.B.P.	2.207	0.615	7.924	0.225
≥1.5 million L.B.P.	0.589	0.213	1.627	0.307
≤300 USD	0.459	0.160	1.320	0.149
More than 300 USD	0.096	0.032	0.284	<0.001

## Data Availability

All the study data are reported in this paper.

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
