# Peer review of "The Ukraine–Russia War Is Deepening Food Insecurity, Unhealthy Dietary Patterns and the Lack of Dietary Diversity in Lebanon: Prevalence, Correlates and Findings from a National Cross-Sectional Study"

_nutrients, 2022, doi:10.3390/nu14173504_

Round 1

Reviewer 1 Report

In this very interesting cross-sectional study, the authors examined food diversity and unhealthy dietary patterns among Lebanese household’s members. This is a very important topic. The authors found that food insecurity was highly prevalent in Lebanon, and they also identified factors related to food insecurity. There are several main suggestions:

1.       Add a flow chart to illustrate the study population recruitment process. This will help readers better understand how the study was conducted and why the study population has good sampling representativeness.

2.       The authors can further discuss the short-term and long-term impacts of food insecurity. For example, there are immediate demographic impacts and long-term health impacts of undernutrition. The relevance of these impacts should be discussed. These include but should not limit to: Li & Lumey, Nutrients 2022; Rudnytskyi et al., Canadian Studies in Population, 2015.

There are some minor issues:

1.       In section 3.1, the authors can divide the paragraph into two or three paragraphs. Otherwise, this part should be condensed.

2.       Line 82, I think that it may be better to use past tense.

Reviewer 2 Report

This manuscript reports on a study exploring food insecurity and diet diversity in Lebanon in mid 2022. Finding that thanks to the Russia-Ukraine war food security and diet diversity have diminished. In general, this is a well-considered manuscript and with a few changes could be publishable.

For an international readership, it would be useful to include some background or context to why Lebanon is experiencing such hardship – this could be included in the paragraph starting line 70. Some of this comes out in the discussion, but I think the introduction would benefit more from this context.

You say that this is a representative sample, but you have not provided any power or sample size calculations. This is a requirement for such a statement.

You should say how the survey was validated. Is there a reference for this? Has the survey been previously published? This should also be included.

Results presented line 251: I don’t  think you can ascribe your findings to remarkable changes. This is a snapshot in time, not a pre/post study. This language should be changed to reflect this.

I would re-strucutre the results so that the food insecurity data are presented after the demographic material as this is what your study is really about

A few minor editorial points

In the abstract, line 29 says data was, this should be data were (and again line 120)

In the abstract, you don’t need both numbers and structured headings, eg. (1) Background

Line 44, remove ‘The’ at the start of the sentence

There are some tense problems in the introduction

Throughout, consider removing some of the sentence joiners (moreover, furthermore, however, consequently). You don’t need these to start every sentence.

Avoid conjunctions (line 159, didn’t should be did not)

Author Response

Reviewer 2

This manuscript reports on a study exploring food insecurity and diet diversity in Lebanon in amid 2022. Finding that thanks to the Russia-Ukraine war food security and diet diversity have diminished. In general, this is a well-considered manuscript and with a few changes could be publishable.

For an international readership, it would be useful to include some background or context to why Lebanon is experiencing such hardship – this could be included in the paragraph starting line 70. Some of this comes out in the discussion, but I think the introduction would benefit more from this context.

Reply: Thank you for this meaningful hint, the introduction is modified as suggested.

You say that this is a representative sample, but you have not provided any power or sample size calculations. This is a requirement for such a statement.

Reply: Thank you for this comment, this issue is managed and the information concerning calculation of representative sample size was added to the paragraph “study design and sampling” and was added to the flow diagram showing the recruitment process.

You should say how the survey was validated. Is there a reference for this? Has the survey been previously published? This should also be included.

Reply: Thank you for raising this comment. The questionnaire used in this study was a collection between valid questionnaires published previously:

  1. The food security questionnaire called :the Arab Family Food Security Scale (AFFS) that was used to assess food insecurity among households was derived from: ‌Sahyoun NR, Nord M, Sassine AJ, Seyfert K, Hwalla N, Ghattas H. Development and Validation of an Arab Family Food Security Scale. The Journal of Nutrition. 2014 Mar 5;144(5):751–7.
  2. Information concerning socioeconomic and sociodemographic factors along with the calculation of the food consumption score were derived from : VAM Resource Centre. FCS—Food Consumption Score. Available online: https://resources.vam.wfp.org/data-analysis/quantitative/food-security/fcs-food-consumption-score (accessed on 22 July 2022)
  3. As for the food-related behaviors and food consumption patterns, our research group published previously these questionnaires:
    1. Hoteit, M., Mortada, H., Al-Jawaldeh, A., Mansour, R., Yazbeck, B., &; The Regional CORONA COOKING Survey Group. (2022). Dietary Diversity in the Eastern Mediterranean Region Before and During the COVID-19 Pandemic: Disparities, Challenges, and Mitigation Measures. Frontiers. https://doi.org/10.3389/fnut.2022.813154
    2. Hoteit M, Al-Atat Y, Joumaa H, Ghali SE, Mansour R, Mhanna R, et al. Exploring the Impact of Crises on Food Security in Lebanon: Results from a National Cross-Sectional Study. Sustainability. 2021 13(16):8753. https://www.mdpi.com/2071-1050/13/16/8753/htm#B7-sustainability-13-08753
    3. Hoteit M, Mortada H, Al-Jawaldeh A et al.COVID-19 home isolation and food consumption patterns: Investigating the correlates of poor dietary diversity in Lebanon: a cross-sectional study [version 1; peer review: 2 approved]. F1000Research 2022, 11:110 (https://doi.org/10.12688/f1000research.75761.1

Results presented line 251: I don’t think you can ascribe your findings to remarkable changes. This is a snapshot in time, not a pre/post study. This language should be changed to reflect this.

Reply: the authors would like to thank the reviewer for raising this comment. To clarify, only the questions regarding food consumption data and food dietary behaviors were asked to investigate changes from the pre-war time to the war time.  We wanted to highlight the impact of the war itself on the changes in these behaviors.

Moreover, we already published data on dietary diversity (Hoteit, M., Mortada, H., Al-Jawaldeh, A., Mansour, R., Yazbeck, B., &; The Regional CORONA COOKING Survey Group. (2022). Dietary Diversity in the Eastern Mediterranean Region Before and During the COVID-19 Pandemic: Disparities, Challenges, and Mitigation Measures. Frontiers. https://doi.org/10.3389/fnut.2022.813154) and  food insecurity (Hoteit M, Al-Atat Y, Joumaa H, Ghali SE, Mansour R, Mhanna R, et al. Exploring the Impact of Crises on Food Security in Lebanon: Results from a National Cross-Sectional Study. Sustainability. 2021 13(16):8753. https://www.mdpi.com/2071-1050/13/16/8753/htm#B7-sustainability-13-08753), that described the food security and dietary diversity situation analysis before the war. These references are available in the manuscript and were compared to the current findings.

 This issue was resolved and the below text was added in line 207: 

 “The second part consisted of 10 questions concerning the impact of the Russia-Ukraine War on food-related habits, including food purchasing behaviors, food consumption habits, and food storage These questions were derived from previous published paper by other authors [21] and by our research group describing the same study variables before Russia-Ukraine war [6]. It asked about the changes in dietary patterns and in the consumption data by households during the current war period in comparison to the preceding one”.

Also, the data showing the changes (pre-war versus current period) was amended in the manuscript accordingly.

I would re-structure the results so that the food insecurity data are presented after the demographic material as this is what your study is really about.

Reply: Thank you for raising this comment, results were presented as suggested.

A few minor editorial points

In the abstract, line 29 says data was, this should be data were (and again line 120)

In the abstract, you don’t need both numbers and structured headings, eg. (1) Background

Line 44, remove ‘The’ at the start of the sentence

There are some tense problems in the introduction

Throughout, consider removing some of the sentence joiners (moreover, furthermore, however, consequently). You don’t need these to start every sentence.

Avoid conjunctions (line 159, didn’t should be did not)

Reply: All editorial points were reviewed and corrected.

Round 2

Reviewer 2 Report

Great work on addressing these comments. Just fix up figure one, as this seems to have been corrupted.